# ChatLogic: Integrating Logic Programming with Large Language Models for Multi-Step Reasoning

**Zhongsheng Wang, Jiamou Liu*, Qiming Bao, Hongfei Rong, Jingfeng Zhang**

School of Computer Science, University of Auckland

zwan516@aucklanduni.ac.nz, jiamou.liu@auckland.ac.nz, {qbao775, hron635}@aucklanduni.ac.nz,
jingfeng.zhang@auckland.ac.nz

## Abstract

Large language models (LLMs) like ChatGPT and GPT-4 exhibit impressive capabilities in a wide range of generative tasks. However, their performance is often hindered by limitations in accessing and leveraging long-term memories, leading to specific vulnerabilities and biases, especially in prolonged interactions. This paper introduces ChatLogic, an innovative framework that augments LLMs with logical reasoning. In ChatLogic, LLMs play a central role, acting as the controller and engaging in every phase of the system's operation. We present a novel method for translating logical questions into symbols integrated with a reasoning engine. This approach harnesses the contextual understanding and mimicking skills of LLMs, employing symbolic memory to enhance multi-step deductive reasoning capabilities. Our findings reveal that the ChatLogic framework markedly improves the multi-step reasoning capabilities of native LLMs. The source code and data are available at https://github.com/Strong-AI-Lab/ChatLogic.

## Introduction

Recent advancements in large language models (LLMs) such as ChatGPT-3.5, GPT-4 (OpenAI 2023), and Llama2 (Touvron et al. 2023) have significantly enhanced their capabilities in various industries, proving invaluable in solving complex real-world problems. These models are revolutionizing sectors like customer service, healthcare, and education through their nuanced contextual comprehension and advanced conversational abilities. However, when it comes to multi-step logic reasoning tasks, LLMs face notable challenges.

While these models excel in content generation, they struggle with consistently producing coherent responses in tasks that require multi-step reasoning. Their training methodology, primarily based on the 'next-token prediction' approach, limits their ability to apply logical rules and deep contextual understanding essential for such tasks. For example, Figure 1 shows how to let LLMs find a reasonable explanation path as the judgment result of the problem in the known randomly disrupted proposition sequence. This represents a critical area for improvement in current LLMs.

---

*Corresponding author
Copyright © 2024, Association for the Advancement of Artificial Intelligence (www.aaai.org). All rights reserved.

Further complicating this issue is the inherent token limitation of LLMs, which becomes apparent in continual dialogues (Thirunavukarasu et al. 2023). The token caps in models like GPT-3.5 and GPT-4, while extendable through engineering prompts or technologies like Recursive Model Training (Bulatov, Kuratov, and Burtsev 2023), still pose a significant constraint. This limitation is particularly pronounced in multi-turn conversations, a common feature in multi-step logic reasoning tasks.

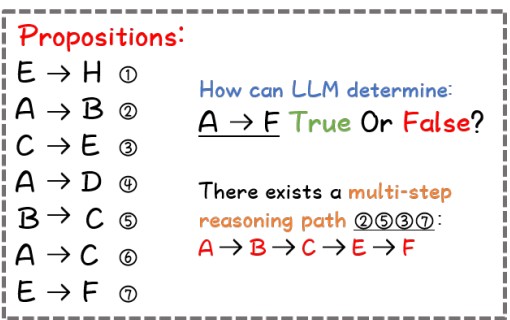

Figure 1: Demo illustrating how LLMs can effectively identify and follow correct and logical reasoning paths to solve complex multi-step reasoning problems. In this instance, our objective is to let LLMs recognize the presence of an established path, ABCEF, thereby enabling them to accurately deduce that the statement 'A infers F' is true.

To address these limitations, innovative approaches such as external memory augmentation are being explored (Zhong, Lei, and Chen 2022). This method involves integrating LLMs with extensive databases, aiming to enhance their reasoning capabilities (Borgeaud et al. 2022). However, this integration brings its own set of challenges, such as the potential embedding of biases from the retrieval models into the LLMs (Khattab et al. 2022), which could affect their accuracy and stability.

Our work introduces ChatLogic, a framework that augments LLMs with a logical reasoning engine to enhance their inferential capabilities. In this framework, we have innovatively implemented a 'Mix-shot Chain of Thought' technique. This approach significantly enhances the performance of LLMs by combining various prompt en-

gineering methods. Mix-shot CoT efficiently guides the model through logical reasoning steps, achieving improved problem-solving with minimal resource consumption. ChatLogic is designed to be compatible with existing LLMs, significantly increasing their accuracy, especially in high-precision scenarios. The framework orchestrates the functioning of an LLM, enabling it to efficiently generate responses across various tasks.

At the heart of ChatLogic is the transformation of natural language into logical symbols, a process executed through pyDatalog. The primary objective of ChatLogic is to reinforce the stability of the reasoning process, ensuring that LLMs can handle intricate reasoning tasks with enhanced reliability and precision. The main characteristics of our framework are summarized below:

- ChatLogic, by combining LLMs with pyDatalog, translates natural language queries into logic programs, enhancing inference accuracy. This improvement is notably evident in multi-step reasoning tasks, as demonstrated on datasets such as PARARULE-Plus[1], CONCEPTRULES V1, and CONCEPTRULES V2.

- ChatLogic mitigates information loss, making it effective in addressing the long sequence limitation prevalent in the adoption of LLMs for multi-step reasoning tasks.

- ChatLogic incorporates automated enhancements for logic program execution, including a syntax correction module. This module refines a logic program by learning from previous executions, significantly improving the practical application and effectiveness of the generated code.

## Related Work

### LLMs Reasoning:

LLMs are considered to have reasoning abilities similar to human cognition (Huang and Chang 2022). Despite facing challenges in multi-step logical tasks involving contemporary information or complex logical sequences (Creswell, Shanahan, and Higgins 2022), emerging approaches like self-consistency (Wang et al. 2022) show promise in enhancing performance, particularly in areas such as arithmetic and common sense reasoning. The effectiveness of causal reasoning pathways (Creswell and Shanahan 2022) is also crucial, ensuring that the output of LLMs is not only accurate but also transparent and verifiable. However, the most impactful method is the Chain of Thought (CoT) (Wei et al. 2022), which reveals the intermediate reasoning steps used by these models in problem-solving, allowing for continual self-correction of logical thinking, thereby greatly enhancing the rationality of their reasoning capabilities.

### LLMs Code Generation:

LLMs have demonstrated the ability to generate code in various programming languages to meet users' specific needs (Yang, Ishay, and Lee 2023). However, how to directly apply the generated code to actual environments remains an

[1]https://huggingface.co/datasets/qbao775/PARARULE-Plus

issue to be resolved. The LOGIC-LM method (Pan et al. 2023) combines LLMs with the principles of symbolic reasoning, clarifying the nuances of text-to-Prolog code conversion by utilizing feedback from symbolic solvers during self-improvement. In terms of optimization, the SELF-DEBUGGING approach (Chen et al. 2023) leads the post-code generation phase. It endows LLMs with the ability to debug their output, reinforcing the theme of continual refinement in the generated code. Even for LLMs that may not fully understand the demonstration samples due to a lack of pyDatalog knowledge in their pre-training data, they can still produce high-precision outputs simply through 'imitation'. Our ultimate goal is to generate code that perfectly meets the requirements and can be directly deployed locally.

### LLMs Prompt Engineering:

Prompt engineering in LLMs functions akin to psychological suggestions, guiding the model towards specific predictions (Wang et al. 2020). Few-shot learning emphasizes training models with the least labeled data for optimized task performance. Notably, models like GPT-3 can handle tasks with a few examples, comparable to fine-tuned models (Brown et al. 2020). Enhanced by prompt engineering, their reasoning capabilities are magnified. Zero-shot Prompting (Reynolds and McDonell 2021) fully relies on the model's vast intrinsic knowledge and training corpus, taking on the entire responsibility of problem-solving. Surprisingly, it often produces results that exceed expectations in many cases, despite limited guidance. Additionally, Zero-shot CoT (Kojima et al. 2022) is also considered to be the best reasoning prompting at present. By using the specific prompt 'Let's think step by step' and the corresponding two-stage prompt technique, significant improvements are achieved in multiple reasoning-related zero-shot tasks, far surpassing previous zero-shot learning.

Although not every situation using Zero-shot CoT results in optimal content output, in most cases targeting specific downstream tasks, LLMs have shown potential in the current field to effectively perform tasks using small sample techniques and combining external enhancement symbols (Song et al. 2022; Liu et al. 2023). In ChatLogic, we create independent prompt templates for different links in the framework and call them independently. By emphasizing LLMs' inherent reasoning capabilities and combining them with basic symbolic rules, preliminary results suggest a promising direction.

## Task Definition

In our experience with advanced LLMs such as Llama2 and GPT-4, we've noticed their impressive ability to convert text into formal structures like math equations (He-Yueya et al. 2023) and programming languages (Vaithilingam, Zhang, and Glassman 2022). However, these models sometimes struggle with complex, multi-step reasoning tasks. As the reasoning depth increases, the challenge escalates, and LLMs often miss key reasoning steps.

Acknowledging these characteristics, our primary goal is to boost the capability of LLMs to effectively represent

problems in logic programming languages, particularly py-Datalog, which is a Python library that integrates the logic programming paradigm, particularly useful for declarative reasoning and complex querying. It allows for sophisticated rule-based logic and inference to be seamlessly incorporated into Python applications, enhancing their capabilities for decision-making processes. To accomplish this goal, we aim to address the following two subtasks.

**Augmenting the Inferential Abilities of LLMs:** Our preliminary objective is to exploit the one-shot generalization and zero-shot thinking capabilities of LLMs. To accomplish this, we aim to familiarize the model with the intricacies of the symbolic language, specifically using pyDatalog. We introduce them to this language through meticulously crafted examples that cater to all potential edge cases. The union of structured pyDatalog syntax and these detailed examples is paramount in guiding LLMs to comprehend and handle multi-step reasoning.

**Amplifying the Executability of Automated Code Generation Processes:** An LLM's translation from text to code (Budinsky et al. 1996) often isn't flawless on the first attempt and may contain errors. We aim to design a specific module in ChatLogic, to ensure comprehensive and accurate translation of natural language content to code. The produced code should be readily executable locally, directly yielding the desired results.

# ChatLogic

This section provides a detailed overview of the ChatLogic framework, with a particular emphasis on the finer details of its constituent parts and our innovative strategy regarding Mix-shot CoT.

## Framework Overview

The ChatLogic framework comprises four primary phases: input processing, Semantic Correction, Syntax Correction, and local execution response, as meticulously illustrated in Figure 2. The entire process from problem input to result output is depicted in the image in the form of a demonstration. It is worth noting that the initial version of the logic code generated by LLMs at the beginning, after continuous revision through two or more iterations within two modules (semantic and syntactic correction), as a direct code solution for multi-step reasoning Executability is significantly improved, resulting in more precise results. Moreover, the enhanced refinement of this code substantially bolsters its executability, contributing to improved system performance and accuracy.

Algorithm 1 delves deeper into the comprehensive algorithmic process for querying response data in ChatLogic. Apart from the locally executed part, all sub-tasks within ChatLogic are controlled and driven by LLMs acting as components. It consists of two loops, each corresponding to the content of two correction phases. We observe that LLMs excel at semantic corrections, and with a limited number of modifications, correct text translations can be achieved. In lines 5 and 6 of the code, we employed zero-shot CoT to

assist in determining the textual similarity of two propositions. Based on the judgment, we update the 'DifferentFlag' label, which influences the progression of the loop process of 'Semantic Correction'. However, syntax corrections are not always reliable, they may get stuck in an infinite loop, repeatedly performing meaningless tasks. To address this issue, we consider introducing an upper loop limit. Although this somewhat diminishes ChatLogic's inferential capabilities, it significantly enhances the framework's robustness, making it better suited for multi-step deductive reasoning tasks.

## Mix-shot CoT

Our innovative mix-shot CoT (Chain of Thought) approach represents a groundbreaking hybrid methodology, blending the strengths of zero-shot CoT and one-shot learning to create a more versatile and effective learning paradigm for language models. At its core, mix-shot CoT leverages the language model's innate ability for autonomous sub-task identification, a characteristic of zero-shot learning, and enriches it with the precision of one-shot learning through strategically chosen demonstration examples. This dual approach allows for dynamic adaptation to the complexity and specific requirements of various tasks. For tasks demanding high accuracy and nuanced understanding, like complex problem-solving or advanced language interpretation, mix-shot CoT guides the model using high-quality demonstration examples as templates, enhancing its precision and contextual depth. In addition, in scenarios that require more extensive analysis, such as text similarity comparison between generated propositions and original propositions in ChatLogic, LLMs are given greater autonomy to leverage their analytical capabilities and generate innovative solutions, thereby developing their ability to navigate massive amounts of information and generate unique insights.

Moreover, mix-shot CoT is designed to cultivate an adaptable learning process in language models, harmoniously combining structured guidance with the freedom of exploration. This flexibility is crucial in enabling the model to not only accurately follow established patterns but also to innovate and adapt to a diverse range of tasks.

In our comparative analysis, as depicted in Table 1, our mix-shot CoT methodology showcases a considerable leap in performance by striking an optimal balance between the grounded precision of one-shot learning and the generative flexibility of zero-shot CoT. Our approach reduces hallucination significantly and boasts the highest task-specific accuracy due to its judicious use of high-quality demonstration examples. While it does not entirely eliminate the need for demonstrations like zero-shot CoT, nor does it match the minimal hallucination levels of one-shot learning, the mix-shot CoT's enhanced adaptability and efficiency make it a powerful tool in the realm of prompt engineering. By acknowledging the limitations of requiring some demonstrations and not being as inherently scalable as zero-shot CoT, our mix-shot CoT nonetheless stands out for its pragmatic effectiveness in real-world applications where precision and adaptability are paramount.

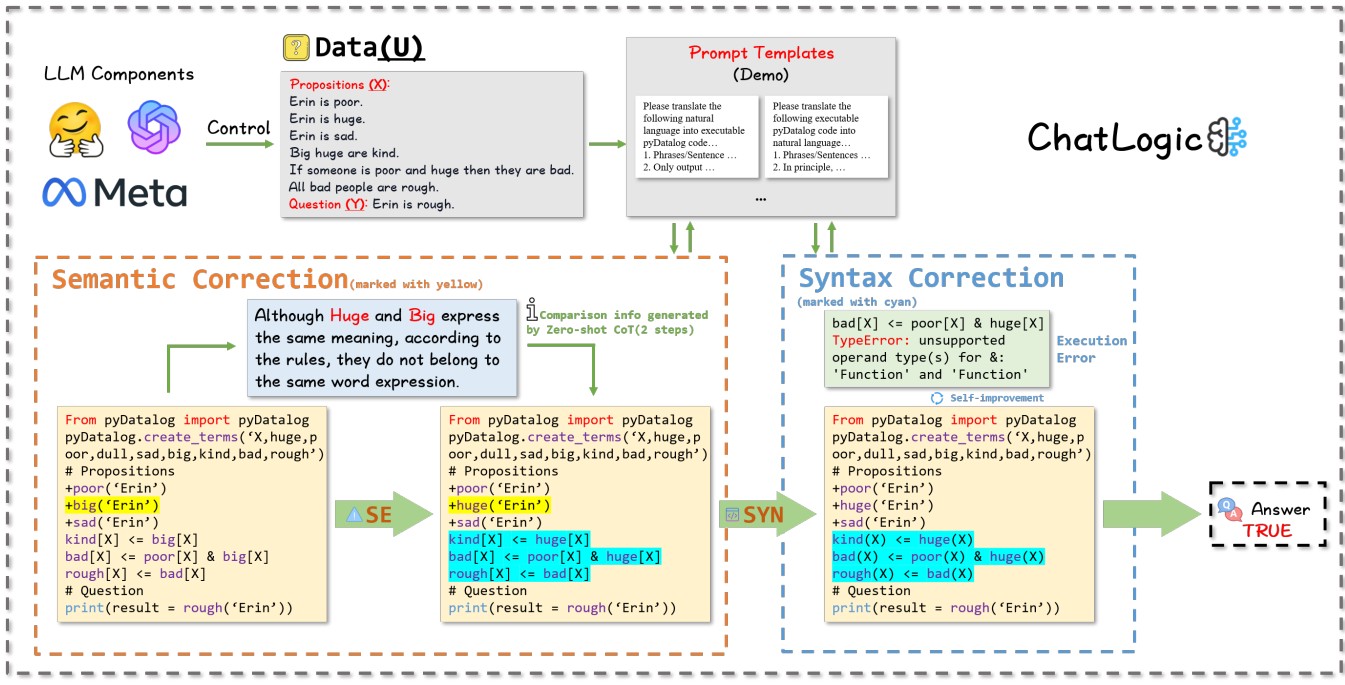

Figure 2: ChatLogic containing more details uses LLMs as controllers, calls appropriate demonstration examples from Prompt Templates, guides the two modules of semantic correction(SE) and syntax correction(SYN) to output correct code, and produces execution results. This is an excerpt of a specific question in PARARULE-Plus and the code generation process. The yellow portion represents the achievements of SE, and the cyan portion represents SYN.

---

**Algorithm 1: The Algorithm of ChatLogic**

---

**Input**: U ← Rules supplemented based on the close-world assumption
X ← Proposition group (contains facts and rules)
Y ← Question
**Output**: TRUE/FALSE (Answer to Y given U, X)
  1: DifferentFlag = TRUE
                                                                    ▷ Semantic Correction
  2: **while** DifferentFlag **do**
  3:     Code ← PropositionTransformation($X, Y, U$)      ▷ Generate logic program based on close-world assumption
  4:     RevProposition ← ReverseTransformation(Code, U)              ▷ Convert code back to natural language
  5:     DifferentInfo ← TextComparison($(X, Y)$, RevProposition)
  6:     DifferentFlag ← JudgeInfo(DifferentInfo)      ▷ Determining semantic similarity status with zero-shot CoT in 2 steps
  7: **end while**
                                                                    ▷ Syntax Correction
  8: ExecutionError = NULL                                          ▷ Execution result record
  9: **while** Code cannot be executed **do**
 10:     Code ← CodeImprovement(Code, $ExecutionError$)              ▷ Improve code based on error info
 11:     **if** Running Time Overflow **then**
 12:         Terminate WHILE Loop
 13:     **end if**
 14: **end while**
 15: **return** CodeExecution(Code)                          ▷ Get results by executing pyDatalog code locally

---

## Evaluation

In this section, we conduct experiments to evaluate the effectiveness of ChatLogic-augmented LLMs. Our experimental results show that **ChatLogic+LLMs** significantly outperform baseline LLMs, highlighting the advantages of using logical symbols to enhance the multi-step reasoning capabilities of LLMs.

Table 1: Comparison among different methods for prompt engineering

| Attributes | Prompt Engineering Methods | | |
|---|---|---|---|
| | One-shot Learning (Brown et al. 2020) | Zero-shot CoT (Kojima et al. 2022) | Mix-shot CoT (Ours) |
| Hallucination | Least hallucination | High hallucination | Less hallucination |
| Accuracy | Poor performance | Relatively high accuracy | Highest accurate response to task |
| Expansibility | Different demos needed for different tasks | No demo needed, only simple guidance required | Fewer demos, and improved performance than fewer shots |

## Datasets and Metrics

All reasoning questions in PARARULE-Plus adhere to the closed world assumption, totaling approximately 400,000 samples. It features linguistic information in two contextual scenarios: People and Animal. For this dataset, we conducted our experiment by randomly selecting 50 instances from each depth level in both the Animal and People categories, combining them to form a set of 100 test cases for each depth level, ranging from *Depth=2* to *Depth=5*.

In addition to PARARULE-Plus, we also incorporate the CONCEPTRULES V1[2] and CONCEPTRULES V2[3] datasets in our study. These datasets contain samples that require multi-step reasoning, with depths up to 3, making them suitable for evaluating models' capabilities in complex reasoning tasks. They are available in both simplified and full versions. For each version of the CONCEPTRULES datasets, we initially consolidated all data from the train, test, and dev sets into a single pool. From this unified dataset, we then randomly sampled 100 instances for our tests.

In our experiments with ChatGPT, GPT-4, and Llama 2-7B, we aimed to establish a baseline for reasoning capabilities of LLMs as documented in the literature(Hu et al. 2023). A significant part of our study was the implementation of ChatLogic, a framework designed to potentially enhance these models' reasoning accuracy. This involved testing configurations like ChatGPT vs. ChatLogic (ChatGPT) across uniform scenarios using instances from the PARARULE-Plus and CONCEPTRULES datasets. The crux of our hypothesis is that if models augmented with ChatLogic demonstrate improved reasoning performance across various difficulty levels compared to their baseline, it would be a strong indicator of ChatLogic's effectiveness. Such results would suggest that ChatLogic could be a valuable addition to the field of artificial intelligence and natural language processing, affirming its utility in advancing the reasoning capabilities of LLMs.

## LLMs Configuration

In the ChatLogic framework invocation, ensuring the controllability of the text is paramount. For ChatGPT and GPT-4, the model invocation versions are respectively "**gpt-3.5-turbo**" and "**gpt-4**", and both have the hyperparameter temperature set to 0 so that we can have more control over what the LLM outputs. For Llama 2-7B, using an NVIDIA GeForce RTX 3090 with 24GB of memory, we utilized the transformer-based version **meta-llama/Llama-2-7b-chat-hf** [4] provided by Huggingface.

## Intermediate Process

Figure 3 provides an illustrative comparison between two approaches for handling the PARARULE-Plus dataset. On the left, we can observe the inference process generated directly by ChatGPT. While it appears initially logical, upon closer examination, it becomes apparent that this inference process contains numerous logical inconsistencies.

In contrast, on the right side of the figure, we showcase the enhanced performance of ChatGPT when coupled with the ChatLogic framework. This presentation includes only a selected excerpt of the final generated code. When executed locally, this integration produces remarkably precise inference results, addressing the logical inconsistencies encountered in the unassisted ChatGPT inference process.

It is worth noting that after experiments, we found that before calling the text data in the PARARULE-Plus data set, LLMs can not complete the missing information for the relevant text content based on the closed-world assumption (Fei and Liu 2016), resulting in a significant decrease in reasoning efficiency. In order to avoid this situation, we built a script to extract the subject and attribute affiliations in the text through Named Entity Recognition (NER) and complete the missing information in the proposition in the format of artificially synthesized natural language. Then use the ChatLogic framework to enhance text reasoning on LLMs. For example, in the '**Rules**' in Figure 3, through observation we find that no sentence can deduce that someone has the attribute of '**big**', so this attribute must appear in the definition of character attributes. If we do not see tasks with '**big**' attributes in '**Facts**', then we default that they do not have '**big**' attributes, which requires us to manually add them to '**Facts**'. This part also belongs to the specific content represented by **U** in Algorithm 1 and Figure 2. The same approach was not utilized for the two distinct versions (simplified & full) of each of the CONCEPTRULES datasets (V1 & V2), as these versions did not require the supplementation of additional information.

## Result

The experimental findings, as illustrated in Tables 2 and 3, unequivocally showcase the ChatLogic framework's signif-

[2]https://bit.ly/3uVemXG
[3]https://bit.ly/3PApIIB
[4]https://huggingface.co/meta-llama/Llama-2-7b-chat-hf

**Propositions:**
**Facts:**
Anne is huge. Anne is strong. Fiona is thin.
Fiona is little. Alan is clever. Dave is rough. Dave is bad.
**Rules:**
If someone is not big then they are rough.
(Continue on the right)

If someone is not sad then they are kind.
If someone is clever then they are quiet.
If someone is quiet and not poor then they are smart.
If someone is rough and not big then they are dull.
If someone is dull then they are small.
**Question:**
**Anne is tiny?**

**ChatGPT:**
To determine whether "Anne is tiny" is correct, let's examine the given propositions step-by-step and follow the chain of reasoning.
From: Anne is huge.
Based on Proposition 8: "If someone is not big then they are rough. **"Since Anne is huge, she is big."** Hence, she is not rough.
We need to determine if Anne can be categorized as small and tiny.
(WRONG! " Huge " is not the same as "big ". We should not infer downward based on " Anne is big ", but based on the closed-world assumption: The proposition: "Anne is big" is not clearly defined, we should think that Anne is not big.)
Therefore, we don't have enough evidence to say that "Anne is tiny" based on the given propositions.
**Answer: 0 (The statement "Anne is tiny" is wrong based on the provided propositions.)**

**ChatLogic:**
```
# Declare the pyDatalog variables
pyDatalog.create_terms('X,huge,strong,little,
rough,clever,sad,kind,quiet,tiny,big […]')
# Define the facts
+huge('Anne')
+strong('Anne')
+little('Fiona') […]
# Define the rules
rough(X) <= ~big(X)
kind(X) <= ~sad(X)
quiet(X) <= clever(X) […]
# Query the knowledge base
result = tiny('Anne')
if result:
    print(1)
else:
    print(0)
```

```
result: 1
Anne is tiny.
```

Figure 3: Comparison based on the PARARULE-Plus dataset shows that while ChatGPT, even with CoT reasoning, often leads to incorrect inferences, the ChatLogic framework (also driven by ChatGPT) in most cases accurately generates pyDatalog code, highlighting its more reliable reasoning proficiency.

icant enhancement of LLMs' performance, surpassing the Baseline with considerably higher accuracy in most scenarios. While native models demonstrate competency in answering straightforward questions, they exhibit limitations in complex multi-step reasoning tasks, leading to reduced accuracy in more challenging questions. In stark contrast, the amalgamation of ChatLogic with LLMs consistently manifests superior accuracy across various levels of question difficulty. This highlights the critical role of augmenting LLMs with logical symbolic operations in multi-step reasoning. By adopting this methodology, we ensure the retention of comprehensive information in natural language, effectively preventing omissions and the accumulation of errors that could compromise reasoning outcomes. Moreover, this approach enhances the transparency of the reasoning process, thereby elevating the credibility and traceability of the results.

In our analysis of the PARARULE-Plus dataset, the ChatLogic framework consistently outperforms the baseline model ('Base') and 'Zero-shot CoT' in most scenarios. Notably, GPT-4, in conjunction with ChatLogic, exhibits exceptional performance on questions of higher complexity (Depth=4 and Depth=5), underscoring ChatLogic's robust capability in handling intricate problems. Regarding Llama 2-7B, despite its weaker baseline performance, it shows sig-

nificant improvement at all depth levels when assisted by ChatLogic. This indicates the framework's versatility in enhancing multi-step reasoning abilities across different models.

When observing the CONCEPTRULES V1 and V2 datasets, a notable shift in GPT-4's performance becomes evident. With Zero-shot CoT, GPT-4 either parallels or slightly surpasses ChatLogic in many cases, particularly in the full version of the CONCEPTRULES V2 dataset. The performance difference between GPT-4 and ChatLogic on these datasets is more nuanced compared to their performance on the PARARULE-Plus dataset. The inherent robustness of the GPT-4 model, likely due to its larger parameter count, already demonstrates formidable capabilities. This finding underscores the future need for more sophisticated datasets to challenge the upper-performance limits of advanced LLMs. Additionally, it's observed that ChatLogic primarily enhances models with a smaller parameter count, by providing appropriate guidance, thereby boosting their performance on the task. This reaffirms the significant value and relevance of our work, especially in optimizing models that are not inherently equipped with extensive computational resources.

Table 2: Accuracy comparison on the PARARULE-Plus dataset (1 for perfect accuracy), including 'Base' and 'Zero-shot CoT' for reference. Our 'ChatLogic' framework generally outperforms others, showcasing its superior effectiveness with LLMs. For each depth level, the results demonstrated the best performance for every model are highlighted in bold.

| Model | Method | Depth=2 | Depth=3 | Depth=4 | Depth=5 | Total |
|---|---|---|---|---|---|---|
| GPT-3.5 | Base | 0.4 | 0.34 | 0.32 | 0.3 | 0.344 |
| | Zero-shot CoT | 0.42 | 0.42 | 0.41 | 0.3 | 0.3875 |
| | ChatLogic | **0.49** | **0.56** | **0.65** | **0.41** | **0.5275** |
| GPT-4 | Base | 0.65 | 0.75 | 0.42 | 0.4 | 0.555 |
| | Zero-shot CoT | **0.72** | 0.72 | 0.62 | **0.7** | 0.69 |
| | ChatLogic | **0.72** | **0.8** | **0.7** | **0.7** | **0.73** |
| Llama 2-7B | Base | 0.11 | 0.06 | 0.01 | 0.01 | 0.0475 |
| | Zero-shot CoT | 0.15 | **0.13** | 0.08 | 0.06 | 0.105 |
| | ChatLogic | **0.2** | **0.13** | **0.22** | **0.18** | **0.1825** |

Table 3: Accuracy comparison on CONCEPTRULES V1 and V2 datasets (1 indicates perfect accuracy), across both simplified and full versions. Notably, GPT-4 with 'Zero-shot CoT' closely matches or occasionally surpasses our 'ChatLogic' framework in performance. Results showing the best performance for each version of the datasets are highlighted in bold.

| Model | Method | CONCEPTRULES V1 | | CONCEPTRULES V2 | |
| | | simplified | full | simplified | full |
|---|---|---|---|---|---|
| GPT-3.5 | Base | 0.57 | 0.55 | 0.5 | 0.51 |
| | Zero-shot CoT | 0.63 | 0.51 | 0.7 | 0.67 |
| | ChatLogic | **0.69** | **0.67** | **0.79** | **0.74** |
| GPT-4 | Base | 0.95 | 0.94 | 0.89 | 0.86 |
| | Zero-shot CoT | **0.96** | **0.97** | **0.95** | **0.94** |
| | ChatLogic | **0.96** | 0.96 | 0.94 | **0.94** |
| Llama 2-7B | Base | 0.32 | 0.29 | 0.31 | 0.24 |
| | Zero-shot CoT | 0.42 | 0.41 | 0.33 | 0.3 |
| | ChatLogic | **0.48** | **0.49** | **0.37** | **0.36** |

Table 4: Test results of code executability across three datasets. Two modules respectively improve the executability of code, Semantic Correction (SE) and Semantic Correction + Syntax Correction (SE+SYN).

| Dataset | Model | Base | SE | SE+SYN |
|---|---|---|---|---|
| CONCEPTRULES V1 | GPT-3.5 | 0.63 | 0.68 | 0.7 |
| | GPT-4 | 0.92 | 0.96 | 0.96 |
| | Llama 2-7B | 0.31 | 0.60 | 0.62 |
| CONCEPTRULES V2 | GPT-3.5 | 0.6 | 0.73 | 0.8 |
| | GPT-4 | 0.92 | 0.93 | 0.95 |
| | Llama 2-7B | 0.33 | 0.52 | 0.53 |
| PARARULE-Plus | GPT-3.5 | 0.26 | 0.5 | 0.62 |
| | GPT-4 | 0.54 | 0.64 | 0.7 |
| | Llama 2-7B | 0.1 | 0.16 | 0.16 |

## Ablation Study

We have introduced two modules aimed at improving code execution from semantic and syntax perspectives. To demonstrate their role in aiding LLMs' multi-step reasoning, we will separately assess how each module affects the code's executability. We anticipate a gradual increase in successful executions, which would validate the effectiveness of our approach.

For the PARARULE-Plus dataset, we increased our data sampling to 100 random samples from the entire dataset, ensuring that the selection was completely shuffled, with no specific order applied to parameters such as 'Depth (2-5)' and 'Pattern (Animal & People)'. For the CONCEPTRULES V1 and V2 datasets, we thoroughly shuffled all data from both the 'simplified' and 'full' versions of each dataset and randomly selected 100 samples from each. To assess the executability rate of the generated code, we deployed it locally, evaluating solely based on the absence of error messages and the correctness of the output content.

The results are presented in Table 4. In our comparison, we examined the enhancements in code execution rate achieved by LLMs with the gradual integration of different modules. Relative to the baseline, both modules demonstrated incremental improvements in execution rates. Notably, despite Llama 2 not utilizing as much code text for pre-training compared to GPT-3.5 and GPT-4, the Syntax Correction module still proved to be valuable. While it didn't lead to a significant increase in execution rate, its contribution to refining code quality is noteworthy. Furthermore, it's important to highlight that GPT-4, due to its extremely advanced capabilities in some subtasks, has seemingly reached a 'performance ceiling' on the current datasets. This suggests that the datasets' limitations may somewhat constrain its potential.

## Limitation and Future Work

Through experimental evaluations on multiple mainstream LLMs, we observed that ChatLogic+LLMs outperformed native LLMs in terms of performance. The impressive

performance demonstrates our work's effectiveness. However, some issues have also been exposed. PARARULE-Plus is based on the closed-world assumption in question-answering data. Additionally, datasets like CONCEPTRULES V1 and V2, also being artificially constructed, lack natural linguistic expression, which may not fully represent real-world complexities. When confronted with more complex sentences in the context of an open-world assumption, importing, integrating, and inferring external knowledge information that is expressed differently still poses challenges. Despite the valuable results from our experiments in enhancing code reliability, it's essential to acknowledge that the optimization module's applicability is currently limited to specific datasets. Future developments should focus on creating adaptable optimization components (Marvie 2005) to address a wider array of scenarios and data sources.

## Acknowledgement

We extend our heartfelt thanks to HouGarden Company for their financial support. Their commitment has been a key factor in the advancement of our research, providing us with the resources needed to achieve our goals.

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
