# OpenReview forum: "ChatLogic: Integrating Logic Programming with Large Language Models for Multi-step Reasoning"
_AAAI.org/2024/Workshop/NuCLeaR — NuCLeaR 2024_

### Official Review · Reviewer_FcUn · 2023-12-07
**This paper talks about multihop reasoning in LLMs using Chatlogic framework**

**Rating:** 7
**Confidence:** 5

**Review:**

I like the idea similar to iterative syllogism as multihop reasoning. The writing could have been more formal to maintain the flow of reading. Overall, it is a novel proposal with a good code demonstration.

---

### Official Review · Reviewer_Nkof · 2023-12-08

**Rating:** 7
**Confidence:** 4

**Review:**

**Paper Summary.**

This paper presents an interactive prompt-engineering technique for Large Language Models, through a system they call ChatLogic. Rather than asking the LLM to give its chain of thought reasoning in English, the system prompts the LLM to output its reasoning in a logic programming language (specifically, pyDatalog).  The system then exploits this by doing light semantic and syntactic corrections on the program via script-run interactive sessions with the LLM.  The result is that doing this consistently improves the LLM's ability to do multi-step reasoning, measured by accuracy on standard multi-step reasoning datasets.

**Novelty.**

There are three core ideas going on in the ChatLogic system:
  1. Rather than prompting an LLM to give its reasoning in English, prompt it to output logic programming code
  2. Use a scripted LLM interactive session to self-correct *semantic errors*
  3. Use a scripted LLM interactive session to self-correct *syntax errors* (until the program is executable)
  4. Use different types of prompt templates (specifically, zero-shot vs one-shot) at different stages

As the authors mention, the closest system to theirs seems to be LOGIC-LM (Pan et al. 2023), which does both (1) and (3) to perform multi-step reasoning.  Note that (Chen et al. 2023) also considers both (2) and (3), but in the context of self-correcting LLM code in general rather than to improve an LLM's multi-step reasoning.  Scanning the prompt engineering literature, I cannot personally evaluate whether (4) is novel.

The novelty of this work is that, compared to LOGIC-LM, ChatLogic additionally uses (2) semantic self-correction to improve multi-step reasoning.  What's especially interesting about their semantic self-correction is that it is done by translating the output code back to natural language.  The system then uses scripted prompts to align the code with its reverse-translation.  As far as I'm aware, this technique is new.

**Quality.**

The LLM+ChatLogic system performs consistently better than the baseline LLM at standard multi-step reasoning tasks.  It also performs better than an LLM with zero-shot chain-of-thought prompting (Kojima et al. 2022).  These are solid preliminary results.  However, since ChatLogic appears to share a similar approach with LOGIC-LM, it would be good to see how these two systems compare.  The authors also claim in Table 1 that ChatLogic's mix-shot approach is more accurate than one-shot prompting, but they did not include one-shot prompting in their results --- perhaps because it is already worse than zero-shot prompting, but it should be included nonetheless.

In the Evaluation, and throughout the paper, the authors claim that LLM+ChatLogic significantly outperforms, but there are no confidence intervals or discussion of statistical significance supporting this claim.  Additionally, the claim made on page 3 that "our approach reduces hallucination significantly" is entirely unsubstantiated, since the authors do not measure hallucination (higher accuracy does not necessarily entail fewer hallucinations).

Another strength of the paper is that the authors include an ablation study, showing that both the semantic *and* syntactic self-correction components contribute to the system's performance.  I'm curious why they only test the executability of the code produced rather than the performance of each ablated system.  The authors also put a lot of emphasis on their mix-shot prompting (e.g. in the section "Mix-shot CoT"), but I'm skeptical that this meaningfully contributes to the system's reasoning ability.  I recommend testing zero-shot and one-shot prompts against mix-shot in the ablation study.

**Clarity.**

The overall organization of this paper is quite good.  The authors clearly put a good deal of effort into their figures, which do an excellent job communicating the structure and advantages of ChatLogic.

Although the writing is clear in some places, many paragraphs and sentences are confusing and hard to parse.  Here are a few examples:
- (page 3) "It is worth noting that an initially generated code, following successive revisions within the two modules (Semantic and Syntax Correction) through two or more iterations, enjoys a significantly heightened probability of serving as a direct code solution for multi-step reasoning, yielding precise results."
- (page 3) "Conversely, in scenarios needing broader analysis, such as text comparison, thematic exploration, or creative content generation, the model is granted greater autonomy to utilize its analytical prowess and generate innovative solutions, fostering its capability to navigate vast information and produce unique insights."

Similarly, much of the Related Work is hard to follow.  I urge the authors to do a proper revision (prior to submitting a future version of this to a conference) by 1) breaking up these long run-on sentences and 2) setting up sign-posts at the beginning and end of paragraphs that remind your reader of the subject of the paragraph.

Some other issues of clarity: "CoT" is used before "chain-of-thought" is defined.  It would also be nice to see an example of how basic chain-of-thought is used when introducing the concept.  The authors also do not define zero-shot and one-shot CoT in a clear, outlined way (I had to read the literature in order to understand what the difference was).  Perhaps the authors should also note the terms "zero-shot" and "one-shot" were given specific definitions in the prompt-engineering context, which is *not* the same as their meaning in machine learning (this started with (Brown et al. 2020)).

Also, occasionally the authors dress up their work with flowery speech, e.g.
- (page 1) "we have innovatively implemented a mix-shot CoT which significantly boosts LLM performance with minimal resource usage by utilizing diverse prompt engineering techniques."
- (page 3) "Our innovative mix-shot CoT approach represents a groundbreaking hybrid methodology"
- (page 5) "The impressive performance demonstrates our work's effectiveness"
I believe that passages like these are misleading and obscure the actual contribution of the paper (see 'Novelty' above).

**Significance.**

Performing multi-step reasoning is a key challenge for LLMs.  Interest in prompt-engineering for this purpose has exploded recently, so this paper is certainly timely.  I believe this system, along with its cousin in the literature (Pan et al. 2023), provide a unique and important perspective on how to do this: we ought to prompt LLMs to give their reasoning as a logic program, so that we can do meaningful self-corrections that would otherwise be difficult in natural language.

**Typos and Nits.**
- (pages 1, 2) "continuous" -> "continual"
- (page 2) "Surprisingly, often produces..." has no subject
- (page 3) "Overflow" -> "Overview"
- (page 4) "closed-world assumption principle" -> "closed world assumption"
- (page 5) "NER" -> better to spell out Named Entity Recognition for the reader
- (page 6, Figure 3) "Anny is tiny" -> "Anne is tiny" (in bottom-right corner)
- (page 7) "GPT" -> "GPT-4"
- (page 7) The 'Base' in Tables 2 and 3 is slightly different from the 'Base' in Table 4.  You should clarify this in the caption.

---

### Decision · Program_Chairs · 2023-12-11

Accept